# Association of abnormal pulmonary vasculature on CT scan for COVID-19 infection with decreased diffusion capacity in follow up: A retrospective cohort study

**Daniel Salerno** [1], **Ifeoma Oriaku**[1], **Melinda Darnell**[1], **Maarten Lanclus**[2], **Jan De Backer**[2], **Ben Lavon**[2], **Rohit Gupta**[1], **Fredric Jaffe**[1], **Maria Elena Vega Sanchez**[1], **Victor Kim**[1]*, **on behalf of the Temple University Covid-19 Research Group**[¶]

1 Department of Thoracic Medicine and Surgery, Lewis Katz School of Medicine at Temple University, Philadelphia, Pennsylvania, United States of America, 2 FLUIDDA, Inc., New York, New York, United States of America

¶ Membership of the Temple University Covid-19 Research Group is listed in the Acknowledgments.
* victor.kim@tuhs.temple.edu

**Data Availability Statement:** All relevant data are within the manuscript and its Supporting Information files.

## Abstract

### Background

Coronavirus Disease 2019 (COVID-19) is a respiratory viral illness causing pneumonia and systemic disease. Abnormalities in pulmonary function tests (PFT) after COVID-19 infection have been described. The determinants of these abnormalities are unclear. We hypothesized that inflammatory biomarkers and CT scan parameters at the time of infection would be associated with abnormal gas transfer at short term follow-up.

### Methods

We retrospectively studied subjects who were hospitalized for COVID-19 pneumonia and discharged. Serum inflammatory biomarkers, CT scan and clinical characteristics were assessed. CT images were evaluated by Functional Respiratory Imaging with automated tissue segmentation algorithms of the lungs and pulmonary vasculature. Volumes of the pulmonary vessels that were ≤5mm (BV5), 5-10mm (BV5_10), and ≥10mm (BV10) in cross sectional area were analyzed. Also the amount of opacification on CT (ground glass opacities). PFT were performed 2–3 months after discharge. The diffusion capacity of carbon monoxide (DLCO) was obtained. We divided subjects into those with a DLCO <80% predicted (Low DLCO) and those with a DLCO ≥80% predicted (Normal DLCO).

### Results

38 subjects were included in our cohort. 31 out of 38 (81.6%) subjects had a DLCO<80% predicted. The groups were similar in terms of demographics, body mass index, comorbidities, and smoking status. Hemoglobin, inflammatory biomarkers, spirometry and lung volumes were similar between groups. CT opacification and BV5 were not different between groups, but both Low and Normal DLCO groups had lower BV5 measures compared to

**Funding:** Image analyses were funded by Fluidda's COVID19 consortium. The funders had no role in study design, data collection and analysis, decision to publish, or preparation of the manuscript.

**Competing interests:** Over the past three years, VK reports personal fees from Gala Therapeutics, the ABIM, AstraZeneca, and Boehringer Ingelheim outside the submitted work; ML, JDB, BL have received monies from Fluidda, Inc; IO, MD, RG, FJ, MEVS, DS have no competing interests to declare.

**Abbreviations:** ANOVA, Analysis of variance; ARDS, Acute respiratory distress syndrome; BV5, Pulmonary blood vessels ≤5mm; BV5_10, Pulmonary blood vessels 5-10mm; BV10, Pulmonary blood vessels ≥10mm; CI, Confidence Interval; COVID-19, Coronavirus Disease 2019; CT, Computed Tomography; CTEPH, chronic thromboembolic pulmonary hypertension; CRP, C-reactive protein; DLCO, Diffusion capacity of carbon monoxide; $FEV_1$, Forced expiratory volume in 1 second; FRI, Functional Respiratory Imaging; FVC, Forced vital capacity; IRB, institutional review board; LDH, Lactate Dehydrogenase; NHANES, National Health and Nutrition Examination Survey; OR, Odds ratio; PFT, Pulmonary function tests; R, Pearson product-moment correlation coefficient; SPSS, Statistical Package for the Social sciences; TLC, Total Lung Capacity.

healthy controls. BV5_10 and BV10 measures were higher in the Low DLCO group compared to the normal DLCO group. Both BV5_10 and BV10 in the Low DLCO group were greater compared to healthy controls. BV5_10 was independently associated with DLCO<80% in multivariable logistic regression (OR 1.29, 95% CI 1.01, 1.64). BV10 negatively correlated with DLCO% predicted (r = -0.343, p = 0.035).

## Conclusions

Abnormalities in pulmonary vascular volumes at the time of hospitalization are independently associated with a low DLCO at follow-up. There was no relationship between inflammatory biomarkers during hospitalization and DLCO. Pulmonary vascular abnormalities during hospitalization for COVID-19 may serve as a biomarker for abnormal gas transfer after COVID-19 pneumonia.

## Background

Coronavirus Disease 2019 (COVID-19) is a highly contagious respiratory viral illness causing pneumonia, abnormal coagulation, and systemic disease. As of August 2021, it is responsible for over 39.2 million infections and more than 639,000 deaths in the United States. The spectrum of disease severity is vast, ranging from mild illness to acute respiratory distress syndrome (ARDS), multisystem organ failure, and death. It is characterized by sometimes severe and life-threatening systemic inflammation and immune dysregulation [1].

A hallmark feature of COVID-19 is dyspnea and hypoxemia. Much of the dyspnea can be explained by the degree of pneumonia present on imaging, but not uncommonly there is a disconnect between the degree of hypoxemia and visual extent of abnormal lung parenchyma [2]. There is now broad consensus that diverse vasculocentric processes contribute to the striking and surprisingly refractory hypoxemia observed in COVID-19 patients [3]. It has been shown in an autopsy series that those that died of COVID-19 had widespread thrombosis and microangiopathy of the pulmonary vessels [4] suggesting that pulmonary vascular disease plays an important role in the pathophysiology of hypoxemia in the infected patient. Additionally, an unusually high prevalence of venous thromboembolism has been reported in several studies [5, 6]. Previous study of hospitalized COVID-19 patients demonstrated that abnormalities in pulmonary vascular volumes as measured by automated segmentation algorithm were associated with COVID-19 infection and were prognostic for risk of intubation and death [7–9].

Although COVID-19 causes an acute illness, it is now recognized that illness can be prolonged in up to 62% of those infected [10]. Abnormalities in pulmonary function after COVID-19 infection have been described, including restrictive impairments and reductions in diffusion capacity of carbon monoxide (DLCO) [11–15]. The determinants of these abnormalities are unclear. One study found an association with D-dimer measured during acute illness with abnormal DLCO at follow-up [16]. We hypothesized that inflammatory biomarkers and CT scan measures of pulmonary vascular volumes at the time of infection would be associated with abnormal gas transfer at follow-up.

## Methods

In this retrospective cohort study we studied adult subjects who were hospitalized for COVID-19 pneumonia at a single center (Temple University Hospital, Philadelphia, Pennsylvania,

USA) and then discharged between March to July 2020. We included those with follow-up visits and pulmonary function tests (PFTs) available after hospitalization. Those with pre-existing lung disease (except asthma) or known reduced DLCO prior to COVID-19 infection were excluded from our analysis (See **Fig 1** for CONSORT Diagram). This was done to include those with normal or presumably normal lung function prior to infection in order to better differentiate the effects of COVID-19 pneumonia on post COVID-19 lung function. Serum inflammatory biomarkers (including C-reactive protein [CRP], D-dimer, Ferritin, Fibrinogen, Lactose Dehydrogenase [LDH]), CT scan, and clinical characteristics were assessed during the index hospitalization. Serum inflammatory markers were assessed daily while in the hospital. For the sake of our analysis, admission and peak values of the inflammatory biomarkers were used. Pulmonary function tests, including spirometry, lung volumes, DLCO and 6-minute walk tests were performed at 2–3 months after discharge according to the American Thoracic Society/European Respiratory Society Guidelines [17]. Normal values were calculated in usual fashion using NHANES III normative values [18]. We divided subjects into those with a DLCO <80% predicted (Low DLCO) and those with a DLCO ≥80% predicted (Normal DLCO) based on these follow-up pulmonary function tests. This study was approved by the Institutional Review Board (IRB) at Temple University (Protocol Number 26984). All data were fully anonymized before being accessed and the IRB waived the requirement for informed consent.

## Imaging

Because the CT scans were acquired in the course of clinical care and without a standardized protocol, slice thickness varied between 0.625 mm and 3.0 mm. The methods of CT analysis of the pulmonary vascular volumes have been previously described [7]. Briefly, CT images at the time of hospitalization (hospital day 0 or 1) were evaluated by Functional Respiratory Imaging (FRI; FLUIDDA, NV, Belgium) which uses (deep learning trained) automated tissue segmentation algorithms to produce quantitative measures of pulmonary tissue. Measures of the pulmonary vascular volumes include the total vascular volume and the vascular volume of the pulmonary blood vessels grouped using two thresholds, ultimately resulting in three categories: vessels between 1.25 and 5 $mm^2$ in cross sectional area (BV5), vessels bigger than 5 $mm^2$ and less than 10 $mm^2$ in cross sectional area (BV5_10), and vessels between 10 and 100 $mm^2$ in cross sectional area (BV10). These volumes are the combined volumes of the macroscopic intrapulmonary arteries and veins, excluding some of the largest segments which are often obscured in patients with significant pulmonary opacities on CT. Additionally, the amount of opacification on CT was quantified using a deep-learning algorithm trained to quantify the total extent of consolidation, ground glass opacity, edema, reticular disease and crazy paving.

## Statistics

All statistical analyses were performed using SPSS v26 (IBM, Armonk, New York, USA). The two DLCO groups were compared with either unpaired t tests or chi squared tests for continuous or categorical variables, respectively. Levene's test was used to assess equality of variances for the independent variables of interest between the two groups. Fisher's exact tests were used for categorical variables if counts per group were less than 5. Mann Whitney U tests were conducted on non-normally distributed continuous data. Vascular volumes were compared between the low DLCO group, normal DLCO group, and a previously analyzed cohort of 107 healthy controls without COVID-19 infection [7] using one way ANOVA with post hoc analysis for multiple comparisons using Bonferroni tests. Correlations between vascular volumes and DLCO% predicted were calculated with Pearson's test. Multivariable logistic regression

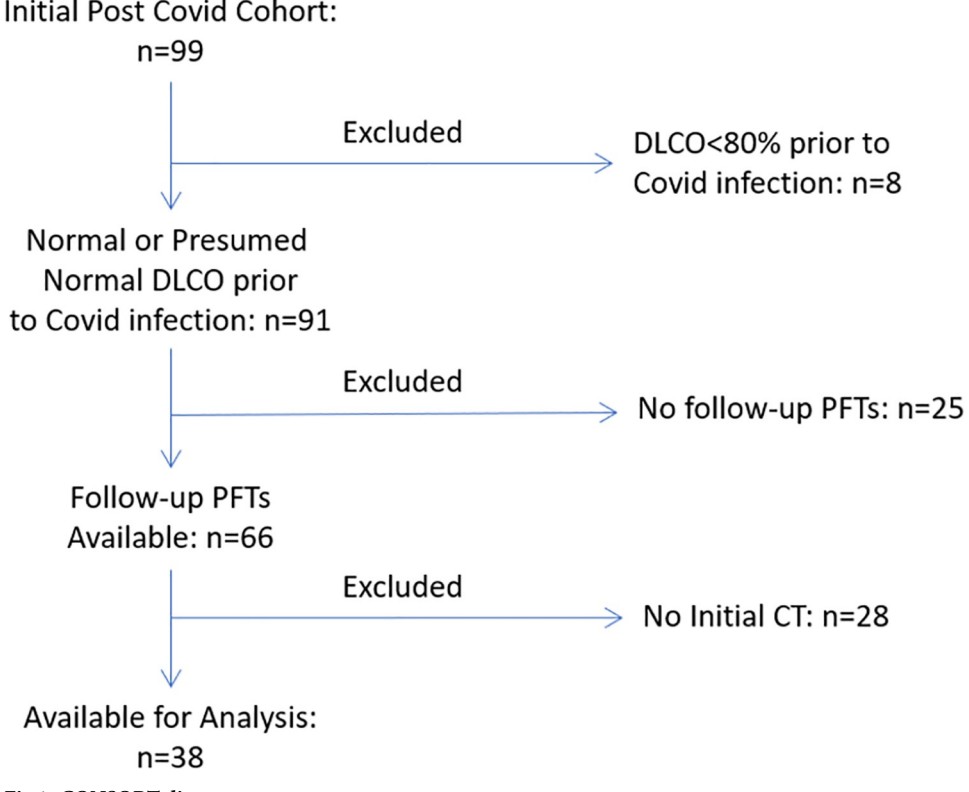

**Fig 1. CONSORT diagram.**

for DLCO<80% predicted was performed with vascular volumes (individual vascular volumes tested in separate models to reduce collinearity) as the independent variables of interest with demographics, smoking status, lung volumes and hemoglobin as covariates. A p value of <0.05 was considered statistically significant.

## Results

Baseline subject characteristics, medical therapies in the hospital and follow-up pulmonary function tests are presented in **Table 1**. There were 12 subjects from the initial cohort that had pulmonary function tests prior to Covid infection, and 8 were excluded due to low DLCO. There were 38 subjects who had pulmonary function tests during follow-up and CT scans on admission. 31 (81.6%) of the cohort had low DLCO, and 7 had normal DLCO. The groups were similar in terms of demographics, body mass index, height, smoking status, and comorbidities. Pulmonary function tests were performed 76.5±35.1 days after the first day of hospitalization. Spirometric measures (FEV₁, FVC, FEV₁/FVC) and lung volumes (TLC) were similar between groups. Both DLCO and DLCO adjusted for alveolar volume (DLCO/VA) were lower in the Low DLCO group.

Serum inflammatory markers, medical therapies, oxygen requirements and needs for respiratory support, and CT measures of opacification and blood volumes are presented in **Table 2**. Of note, admission hemoglobin and peak levels of ferritin, C-reactive protein, LDH, D-dimer, and fibrinogen were similar between the two groups. Peak oxygen requirements were also similar between groups. More patients in the Low DLCO group required advanced respiratory support (high flow nasal cannula or invasive mechanical ventilation) but the difference between groups was not statistically significant. The number of patients in each group

**Table 1. Baseline characteristics and follow-up PFTs.**

| | Normal DLCO | Low DLCO | p |
|---|---|---|---|
| | **n = 7** | **n = 31** | |
| Age (years) | 55.1±15.9 | 59.8±12.4 | 0.399 |
| Gender (male) | 3 (42.9) | 16 (51.6) | 0.676 |
| Race | | | 0.292 |
| Caucasian | 0 (0) | 3 (9.7) | |
| African American | 2 (28.6) | 18 (58.1) | |
| Hispanic | 4 (57.1) | 8 (25.8) | |
| Other | 1 (14.3) | 2 (6.4) | |
| Height (inches) | 64.3±2.4 | 66.6±3.9 | 0.138 |
| BMI (kg/m$^2$) | 39.2±9.5 | 33.9±5.6 | 0.190 |
| Smoking Status | | | 0.585 |
| Nonsmoker | 5 (71.4) | 18 (57.6) | |
| Current Smoker | 0 (0) | 4 (12.9) | |
| Former Smoker | 2 (28.6) | 9 (29.0) | |
| HTN | 4 (57.1) | 22 (71.0) | 0.656 |
| DM | 4 (57.1) | 13 (41.9) | 0.678 |
| CKD | 1 (14.3) | 4 (12.9) | 1.000 |
| CAD | 2 (28.6) | 3 (9.7) | 0.223 |
| CHF | 0 (0) | 9 (27.9) | 0.164 |
| ESRD | 0 (0) | 1 (3.2) | 1.000 |
| Asthma | 0 (0) | 5 (16.1) | 0.561 |
| *Follow Up* | | | |
| Days after Hosp | 74.4±41.8 | 74.9±34.5 | 0.997 |
| mMRC dyspnea score | 0.86±0.90 | 1.40±1.13 | 0.246 |
| 6MWD (feet) | 984±206 | 960±249 | 0.834 |
| FEV$_1$ (%predicted) | 88.4±13.3 | 84.7±19.9 | 0.644 |
| FVC (%predicted) | 88.0±15.2 | 85.2±19.2 | 0.724 |
| FEV$_1$/FVC | 79.7±6.9 | 77.8±6.9 | 0.522 |
| TLC (%predicted) | 80.3±15.0 | 81.1±15.1 | 0.534 |
| DLCO (%predicted) | 89.9±12.6 | 55.5±12.2 | <**0.0001** |
| DLCO/VA (%predicted) | 111.7±13.1 | 82.6±17.4 | <**0.0001** |

Data presented as mean±SD for continuous variables and number (%) for categorical variables. Definition of Abbreviations: DLCO = diffusion capacity of carbon monoxide; BMI = body mass index; HTN = hypertension; DM = diabetes mellitus; CKD = chronic kidney disease; CAD = coronary artery disease; CHF = congestive heart failure; ESRD = end stage renal disease; mMRC = modified Medical Research Council; 6MWD = 6-minute walk distance; FEV$_1$ = forced expiratory volume in 1 second; FVC = forced vital capacity; TLC = total lung capacity; DLCO/VA = DLCO adjusted for alveolar volume.

receiving various medical therapies, including corticosteroids, remdesivir, and monoclonal antibodies, were similar. All patients received at least low dose low molecular weight heparin for venous thromboembolism prophylaxis. Length of stay was similar between groups. 2 patients were diagnosed with a pulmonary embolism and 1 patient with a lower extremity deep venous thrombosis in the Low DLCO group, but none were diagnosed with venous thromboembolism in the Normal DLCO group. However, the incidence of venous thrombo-embolism was not statistically different.

A visual representation of blood vessels color-coded according to size shows the striking difference between a patient that will have a normal diffusion compared to one that will have

**Table 2. Group characteristics of hospitalization.**

|  | Normal DLCO | Low DLCO | p |
|---|---|---|---|
|  | n = 7 | n = 31 |  |
| *Labs* |  |  |  |
| Admit Hgb (g/dL) | 13.6±0.99 | 12.87±1.63 | 0.292 |
| Peak Ferritin (ng/mL) | 247 (226) | 337 (3092) | 0.138 |
| Peak CRP (mg/dL) | 8.4±6.5 | 7.1±5.0 | 0.597 |
| Peak LDH (U/L) | 242±58 | 348±149 | 0.075 |
| Peak D-dimer (ng/mL) | 479 (547) | 663 (46873) | 0.268 |
| Peak Fibrinogen (mg/dL) | 646±173 | 486±189 | 0.267 |
| *Medical Therapies* |  |  |  |
| Corticosteroids | 1 (14.3) | 12 (38.7) | 0.115 |
| Remdesivir | 1 (14.3) | 7 (22.6) | 0.255 |
| Anakinra | 0 (0) | 1 (3.2) | 0.451 |
| IVIG | 0 (0) | 1 (3.2) | 0.451 |
| Sarulimab | 0 (0) | 5 (16.1) | 0.290 |
| Tocilizumab | 0 (0) | 4 (12.9) | 0.327 |
| Gimsilumab | 0 (0) | 1 (3.2) | 0.451 |
| *CT Measures* |  |  |  |
| BV5 (mL) | 79.9±39.9 | 81.0±39.2 | 0.943 |
| BV5_10 (mL) | 42.6±6.9 | 54.2±13.3 | **0.005** |
| BV10 (mL) | 93.0±33.4 | 138.1±53.1 | **0.001** |
| Opacification (mL) | 64.8 (247.6) | 129.2 (1353.2) | 0.218 |
| *Hospital Characteristics* |  |  |  |
| LOS (days) | 4.43±1.51 | 6.26±5.96 | 0.429 |
| Peak O2 Req (L/min) | 0.57±0.98 | 5.66±12.55 | 0.433 |
| Room Air | 5 (71.4) | 17 (54.8) | 0.650 |
| NC O2 | 2 (28.6) | 8 (25.8) |  |
| HFNC O2 | 0 (0) | 4 (12.9) |  |
| IMV | 0 (0) | 2 (6.5) |  |
| New PE | 0 (0) | 2 (6.5) | 1.000 |
| New DVT | 0 (0) | 1 (3.2) | 1.000 |

Data presented as mean±SD or median (IQR) for continuous variables and number (%) for categorical variables.
Definition of Abbreviations: DLCO = diffusion capacity of carbon monoxide; Hgb = hemoglobin; CRP = C-reactive protein; LDH = lactose dehydrogenase; IVIG = intravenous immunoglobulin; LOS = length of stay; NC = nasal cannula; HFNC = high flow nasal cannula; IMV = invasive mechanical ventilation; PE = pulmonary embolism; DVT = deep venous thrombosis.

low diffusion; a healthy control is seen as well. Note how in panel B (future low diffusion patient) the largest blood vessels, in blue, are comparatively prominent, particularly in lower lobes. See **Fig 2**. The quantity of opacification on CT scan was not different between groups (median (IQR) 129.2 (1353.2) vs. 64.8 (247.6) mL in the Low DLCO group and Normal DLCO group, respectively, p = 0.218). BV5 was not different between Low and Normal DLCO groups (81.0±39.2 vs. 79.9±39.9 mL, p = 0.943), but both were lower than the healthy control group (137.0±24.9 mL, p<0.0001 for both Low and Normal DLCO groups, respectively). See **Fig 3**. However, BV5_10 was greater in the Low DLCO group (54.2±13.3 vs. 42.6±6.9 mL, p = 0.005), and BV10 was also greater in the Low DLCO group (138.1±53.1 vs. 93.0±33.4 mL, p = 0.001). Both BV5_10 and BV10 in the Low DLCO group were greater than the healthy controls

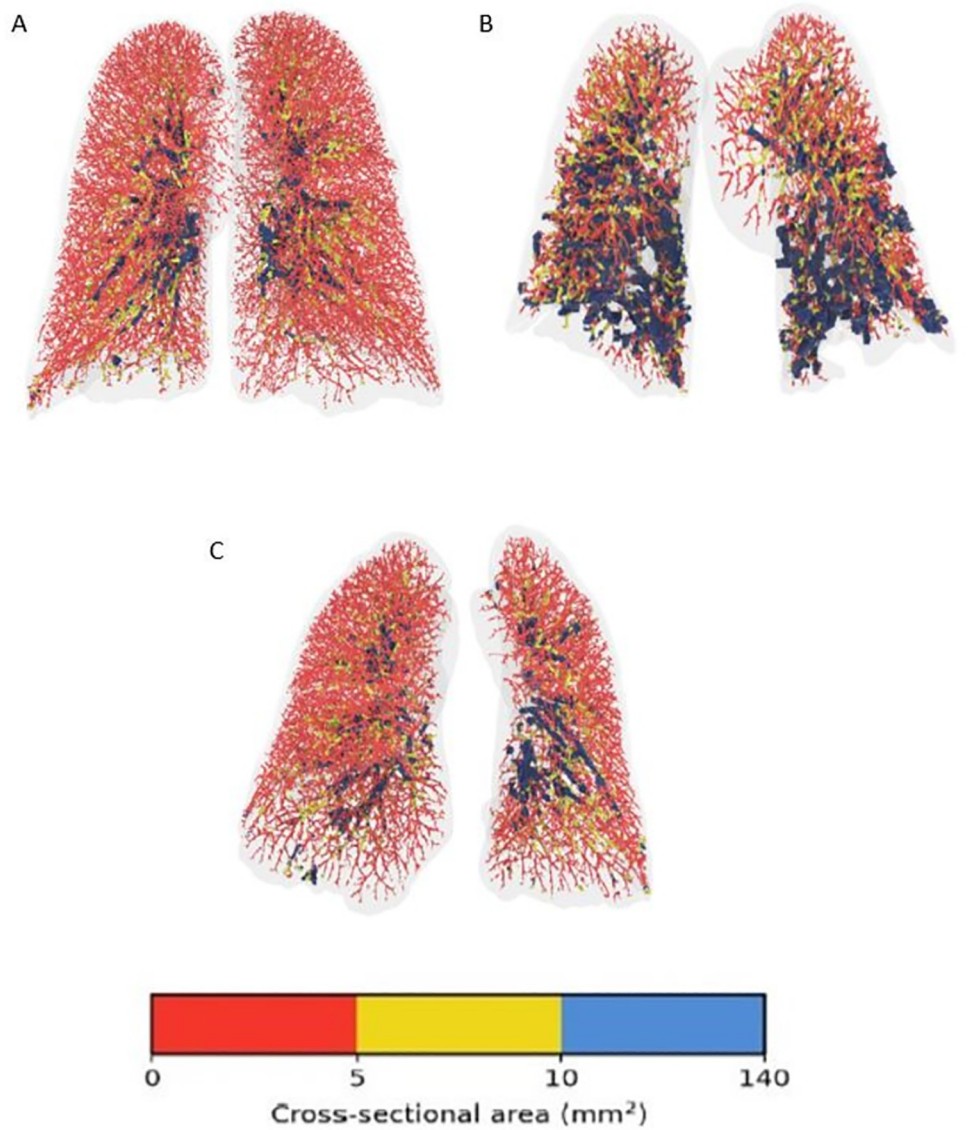

**Fig 2. Pulmonary vascular volumes of 2 COVID-19 patients on hospital admission in comparison to a healthy control.** The patient in Panel A is a healthy control. The patient in Panel B had a DLCO of 32% predicted at follow-up. The patient in Panel C had a DLCO of 117% predicted at follow-up. Blood vessels are colored according to their size. Red denotes the small vessels (5 mm), yellow the mid-size vessels (5–10 mm), and blue the larger vessels (>10 mm).

(p<0.0001 for both). See **Fig 4**. BV5_10 was independently associated with DLCO<80% in multivariable logistic regression (OR 1.29, 95% CI 1.01, 1.64, p = 0.041). BV10 tended to have an independent association with DLCO<80% but it was not statistically significant (OR 1.06, 95% CI 1.00, 1.13, p = 0.072). However, BV10 had a statistically significant negative correlation with DLCO% predicted (r = -0.343, p = 0.035). See **Fig 5**.

## Discussion

We analyzed the impact of COVID-19 hospitalization with pneumonia on follow-up pulmonary function tests. We identified factors during hospitalization that were related to abnormalities in lung diffusion capacity at the time of follow-up. The observed difference in quantified

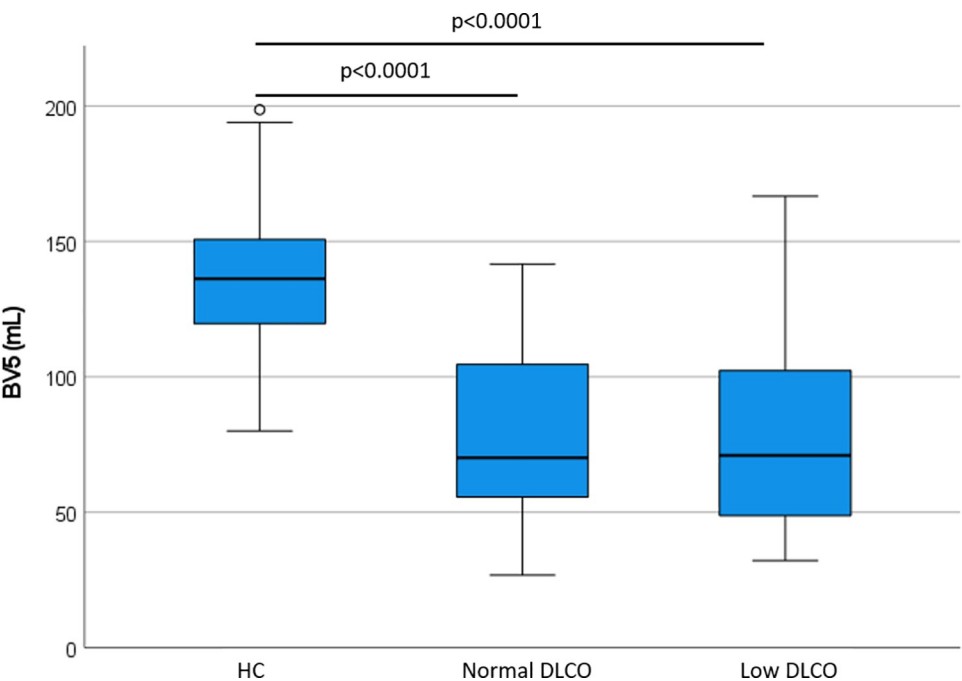

**Fig 3. Pulmonary vascular volumes of pulmonary vessels <5mm².** HC = Healthy Controls.

pulmonary vascular volume distribution and absence of significant restrictive lung disease or anemia in our cohort suggests that the underlying abnormality behind altered respiratory physiology relates to abnormal pulmonary vascular and thrombotic issues rather than persistence or new emergence of pulmonary parenchymal abnormalities. Several reports have indicated that the majority of ground glass opacities clear in a short period of time after COVID 19 pneumonia [19, 20]. A case series of 149 patients [21] did show mostly complete radiological resolution on CT chest 3 weeks after discharge but there was faster clearing of parenchymal abnormalities in younger patients. A more recent prospective observational study from Switzerland [22] on 39 patients found at three months more persistent CT chest abnormalities, also abnormal PFT and decreased quality of life. Also a new editorial about post COVID 19

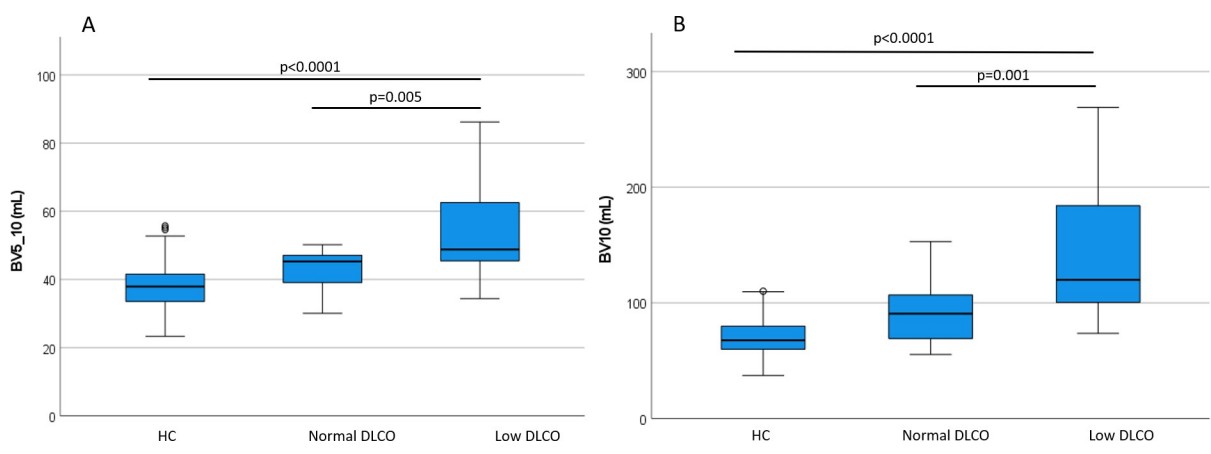

**Fig 4. Pulmonary vascular volumes by DLCO groups.** A) BV5_10 and B) BV10. HC = Healthy Controls.

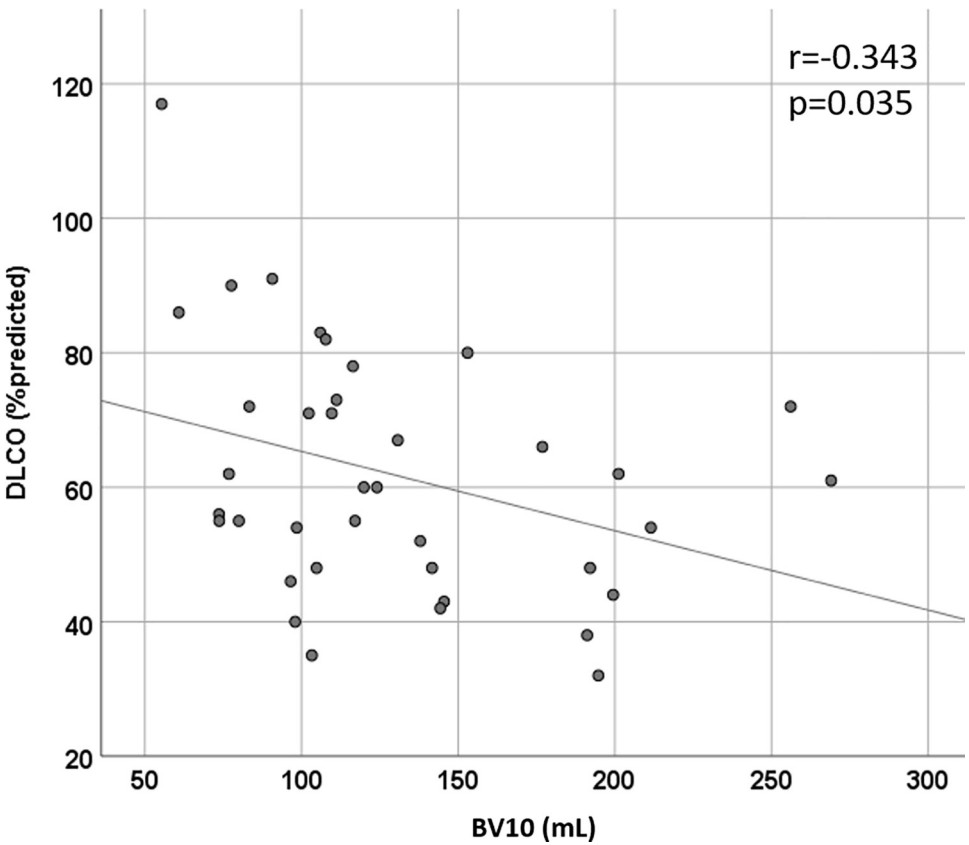

**Fig 5. Correlation between BV10 and DLCO.**

[23] describes three European groups conducting prospective, observational cohorts on patients that were hospitalized with COVID 19; the common denominator was improvement of symptoms, but around half of patients had some persistence of dyspnea and/or abnormal PFT. Those who were in the ICU had more respiratory symptoms and abnormal CT chest at the follow up visit.

Several studies of patients with COVID-19 pneumonia have shown diffusion capacity to be the most common physiologic abnormality at follow-up. A prospective longitudinal study from the Netherlands [15] found decreased diffusion to be the most frequent abnormality in 101 patients. That was also the case in a retrospective series from China [11] and another study from the same country [16]. In a smaller series from Germany with 33 patients [24], decreased diffusion capacity was seen in 77% of patients and it was the only significantly abnormal pulmonary function test that did also include spirometry and lung volumes. Likewise, in 110 patients discharged after COVID-19 pneumonia [12] decreased diffusion was seen in 47.2% of cases, being more common than obstruction or restriction. More recently a systematic review and meta-analysis [25] concluded that after COVID-19 pneumonia the most important component of pulmonary function tests affected was diffusion capacity.

Although we did not find a difference between groups in D-dimer levels, it has been reported elsewhere. The relation between elevation of D-dimer during hospitalization and decreased diffusion capacity in follow-up has been previously described by Zhao et al. [16] in 55 patients. In their multivariable logistic regression model, higher levels of D-dimer at admission were associated with follow up DLCO% predicted <80% (P = 0.031, OR 1.066, 95% CI

1.006 to 1.129). Another study from Spain [26] with a larger cohort found a relationship between maximum D-dimer levels and lower diffusion capacity in subsequent pulmonary function testing. Our report used peak D-dimer during hospitalization instead of D-dimer at admission. D-dimer levels during hospitalization have shown to have relation with mortality, thromboembolic events and need for intubation [27].

Pulmonary vascular abnormalities during COVID 19 pneumonia have been widely reported. Using CT pulmonary angiography and dual energy CT, a retrospective series of 39 mechanical ventilated patients with COVID 19 pneumonia [28] found dilated peripheral vessels (vascular tree-in-bud pattern) and arterial filling defects in the majority of cases. Very similar findings were seen in another retrospective study of 85 patients [29]; their pulmonary dual energy CT angiography revealed a significant number of pulmonary ischemic areas even in the absence of visible pulmonary arterial thrombosis raising the possibility of micro thrombosis associated with COVID-19 pneumonia.

Recent years have seen successful demonstrations of volumetric quantitative CT analysis as a noninvasive means of detecting pulmonary vascular remodeling, a process implicated in a wide of variety of chronic pulmonary conditions Using CT chest for reconstructions of the pulmonary vasculature in patients with chronic thromboembolic pulmonary hypertension (CTEPH) compared to healthy controls there was loss of the distal vessels, dilation of the proximal vessels, and increased vascular tortuosity in those with CTEPH [30]. In a study of smokers [31], loss of the small blood vessels ($<5$ mm$^2$) was found. The magnitude of the changes was correlated to the severity of their lung disease. Correlations between loss of pulmonary vascular volume on QCT analysis and hemodynamics have been demonstrated in patients with known pulmonary hypertension [32].

Regarding the assessment of small pulmonary blood vessels in our cohort, Lins et al. [7] showed previously that patients with COVID-19 pneumonia compared to healthy individuals seem to have redistribution of the blood volume with less vascular volume in blood vessels with an area less than 5mm$^2$ and increased vascular volume in blood vessels with areas of 5 to 10mm$^2$ and more than 10 mm$^2$. This has been commonly interpreted as evidence of dilation of elastic pulmonary arteries proximal to areas of increased pulmonary vascular resistance downstream; the dilation of vessels more proximal than those smaller than 5 mm$^2$ leads to an increase in BV5_10 and BV10 and a reduction in BV5. This is suggestive of persistent dysregulation of pulmonary vascular tone which may contribute to reduced diffusion capacity.

Unique to our data is the fact that these changes are associated with low diffusion capacity in follow-up pulmonary function testing. This has biological plausibility considering the profound changes seen in the lungs obtained during autopsy with patients who died from COVID-19 pneumonia [4]. Changes include severe endothelial injury with disruption of cell membranes, capillary microthrombi and new vessel growth with widespread angiogenesis. Also, in a recent study [33], 83% of patients with severe COVID-19 pneumonia have evidence of intrapulmonary right to left shunt, more evidence of profound vascular anomalies. Our findings support that the presence of pulmonary vascular anomalies in COVID-19 pneumonia patients is associated with future isolated impairment in diffusion capacity.

Our study has several limitations. Firstly, our sample is small and our data was retrospectively collected. Secondly, due to several logistic reasons we could not assess all of our patients at the same time after discharge. Ideally the follow-up pulmonary function tests would have been done at the same time after diagnosis of COVID-19 pneumonia to avoid confounding by the natural history and progression in time of COVID-19 pulmonary disease. Third, the elevation of serum D-dimer levels during hospital admission could have had impact in treatment decisions that might have influenced subsequent pulmonary function testing and diffusion capacity. Fourth, we did not have PFTs prior to hospitalization in the majority of our subjects

to compare to post-hospitalization PFTs. Fifth, it is unknown if and for how long these vascular and PFT changes will persist. Sixth, we did not have in our cohort other test (such as ventilation-perfusions scans, CT pulmonary angiography, dual energy CT) that could have provided more insight into pulmonary vascular injury related to COVID-19 pulmonary disease. Finally, many of the patients hospitalized at our institution were lost to follow-up for a variety of reasons (been unable to contact them, going to long-term care facility after hospitalization, death, etc.), which may have skewed our data towards less sick patients.

## Conclusions

In conclusion, low DLCO is common in hospitalized patients with COVID-19 pneumonia. Unlike other studies, our findings show that inflammatory biomarkers during infection did not relate to DLCO in follow up. However, pulmonary vascular abnormalities, particularly the medium and large vessels, during hospitalization with COVID-19 were related to low DLCO in follow up. Further study regarding the influence of pulmonary vascular volumes on short and long term COVID-19 outcomes is warranted.

## Supporting information

**S1 Data.**
(XLSX)

## Acknowledgments

Temple University COVID-19 Research Group: Aaron Mishkin, Infectious Disease; Abbas Abbas, Thoracic Medicine and Surgery (TMS); Abhijit S. Pathak, Surgery; Abhinav Rastogi, Admin; Adam Diamond, Pharmacy; Aditi Satti, TMS; Adria Simon, Emergency Medicine; Ahmed Soliman, TMS; Alan Braveman, TMS; Albert J. Mamary, TMS; Aloknath Pandya, TMS; Amy Goldberg, Surgery; Amy Kambo, TMS; Andrew Gangemi, TMS; Anjali Vaidya, Cardiology; Ann Davison, TMS; Anuj Basil, Cardiology; Charles T. Bakhos, TMS; Bill Cornwell, TMS; Brianna Sanguily, TMS; Brittany Corso, Internal Medicine; Carla Grabianowski, TMS; Carly Sedlock, Infectious Disease; Catherine Myers, TMS; Chenna Kesava Reddy Mandapati, TMS; Cherie Erkmen, TMS; Chethan Gangireddy, Cardiology; Chih-ru Lin, TMS; Christopher T. Burks, Lab Administration; Claire Raab, Internal Medicine; Deborah Crabbe, Cardiology; Crystal Chen, Internal Medicine; Daniel Edmundowicz, Cardiology; Daniel Sacher, TMS; Daniel Salerno, TMS; Daniele Simon, Emergency Medicine; David Ambrose, TMS; David Ciccolella, TMS; Debra Gillman, TMS; Dolores Fehrle, TMS; Dominic Morano, TMS; Donnalynn Bassler, TMS; Edmund Cronin, Cardiology; Eduardo Dominguez, TMS; Ekamjeet Randhawa, TMS; Eman Hamad, Cardiology; Eneida Male, TMS; Erin Narewski, TMS; Francis Cordova, TMS; Frederic Jaffe, TMS; Frederich Kueppers, TMS; Fusun Dikengil, TMS; Jonathan Galli, TMS; Andrew Gangemi, TMS; Jamie Garfield, TMS; Gayle Jones, TMS; Gennaro Calendo, TMS; Gerard Criner, TMS; Gilbert D'Alonzo, TMS; Ginny Marmolejos, TMS; Matthew Gordon, TMS; Gregory Millio, Internal Medicine; Rohit Gupta, TMS; Gustavo Fernandez, TMS; Hannah Simborio, TMS; John Harwood Scott, TMS; Heidi Shore-Brown, TMS; Hernan Alvarado, Respiratory Care; Ho-Man Yeung, Internal Medicine; Ibraheem Yousef, TMS; Ifeoma Oriaku, TMS; Iris Jung-won Lee, Nephrology; Isaac Whitman, Cardiology; James Brown, TMS; Jamie L. Garfield, TMS; Janpreet Mokha, TMS; Jason Gallagher, School of Pharmacy; Jeffrey Stewart, TMS; Jenna Murray, TMS; Jessica Tang, TMS; Jeyssa Gonzalez, TMS; Jichuan Wu, TMS; Jiji Thomas, TMS; Jim Murrett, Ultrasound Fellow; Joanna Beros, TMS; John M. Travaline, TMS; Jolly Varghese, TMS; Jordan Senchak, Internal Medicine;

Joseph Lambert, TMS; Joseph Ramzy, TMS; Joshua Cooper, Cardiology; Jun Song, Medical Student; Junad Chowdhury, TMS; Kaitlin Kennedy, TMS; Karim B. Ahmed, TMS; Karim Loukmane, TMS; Karthik Shenoy, TMS; Kathleen Brennan, TMS; Keith Johnson, TMS; Kevin Carney, TMS; Kraftin Schreyer, Emergency Medicine; Kristin Criner, Endo; Kumaran, Maruti, Radiology; Lauren Miller, TMS; Laurie Jameson, TMS; Laurie Johnson, TMS; Laurie Kilpatrick, TMS; Lii-Yoong Criner, TMS; Lily Zhang, TMS; Lindsay K. Mcgann, Hospitalist; Llera A. Samuels, TMS; Marc Diamond, TMS; Margaret Kerper, TMS; Maria Vega Sanchez, TMS; Mariola Marcinkienwicz, TMS; Maritza Pedlar, TMS; Mark Aksoy, TMS; Mark Weir, TMS; Marla R. Wolfson, TMS; Marla Wolfson, TMS; Martin Keane, Cardiology; Massa Zantah, TMS; Mathew Zheng, TMS; Matthew Delfiner, Internal Medicine; Matthew Gordon, TMS; Maulin Patel, TMS; Megan Healy, Emergency Medicine; Melinda Darnell, TMS; Melinda Darnell, TMS; Melissa Navaro, TMS; Meredith A. Brisco-Bacik, Cardiology; Michael Bromberg, Hematology; Michael Gannon, Cardiology; Michael Jacobs, TMS; Mira Mandal, TMS; Nanzhou Gou, TMS; Erin Narewski, TMS; Nathaniel Marchetti, TMS; Nathaniel Xander, TMS; Navjot Kaur, TMS; Neil Nadpara, Internal Medicine; Nicole Desai, Internal Medicine; Nicole Mills, TMS; Norihisa Shigemura, Surgery; Ohoud Rehbini, TMS; Oisin O'Corragain, TMS; Omar Sheriff, TMS; Oneida Arosarena, Otolaryngology; Osheen Abramian, TMS; Paige Stanley, TMS; Parag Desai, TMS; Parth Rali, TMS; Patrick Mulhall, Pulm; Pravin Patil, Cardiology; Priju Varghese, Internal Medicine; Puja Dubal, TMS; Puja Patel, TMS; Rachael Blair, TMS; Rajagopalan Rengan, TMS; Rami Alashram, TMS; Randol Hooper, TMS; Rebecca A. Armbruster, Chief Medical Officer; Regina Sheriden, TMS; Robert Marron, TMS; Roberto Caricchio, Rheumatology; Rogers Thomas, TMS; Rohit Gupta, TMS; Rohit Soans, Surgery; Roman Petrov, TMS; Roman Prosniak, TMS; Romulo Fajardo, Surgery; Ruchi Bhutani, TMS; Ryan Townsend, TMS; Sabrina Islam, Cardiology; Samantha Pettigrew, Internal Medicine; Samantha Wallace, TMS; Sameep Sehgal, TMS; Samuel Krachman, TMS; Santosh Dhungana, TMS; Sarah Hoang, TMS; Sean Duffy, TMS; Seema Rani, TMS; Shapiro William, TMS; Sheila Weaver, TMS; Shelu Benny, TMS; Sheril George, TMS; Shuang Sun, TMS; Shubhra Srivastava Malhotra, TMS; Stephanie Brictson, TMS; Stephanie Spivack, Infectious Disease; Stephanie Tittaferrante, Internal Medicine; Stephanie Yerkes, TMS; Stephen Priest, Internal Medicine; Steve Codella, TMS; Steven G. Kelsen, TMS; Steven Houser, Research; Steven Verga, TMS; Sudhir Bolla, TMS; Sudhir Kotnala, TMS; Sunil Karhadkar, Surgery; Sylvia Johnson, TMS; Tahseen Shariff, TMS; Tammy Jacobs, TMS; Thomas Hooper, TMS; Tom Rogers, TMS; Tony S. Reed, Chief Medical Officer; Tse-Shuen Ku, TMS; Uma Sajjan, TMS; Victor Kim, TMS; Whitney Cabey, Emergency Medicine; Wissam Chatila, TMS; Wuyan Li, TMS; Zach Dorey-Stein, TMS; Zachariah Dorey-Stein, TMS; Zachary D. Repanshek, Emergency Medicine.

## Author Contributions

**Conceptualization:** Daniel Salerno, Maarten Lanclus, Jan De Backer, Ben Lavon, Rohit Gupta, Fredric Jaffe, Maria Elena Vega Sanchez, Victor Kim.

**Data curation:** Daniel Salerno, Ifeoma Oriaku, Melinda Darnell, Maarten Lanclus, Jan De Backer, Ben Lavon, Victor Kim.

**Formal analysis:** Maarten Lanclus, Jan De Backer, Ben Lavon, Victor Kim.

**Funding acquisition:** Maarten Lanclus, Jan De Backer, Ben Lavon, Victor Kim.

**Investigation:** Daniel Salerno.

**Methodology:** Daniel Salerno, Jan De Backer.

**Resources:** Rohit Gupta, Fredric Jaffe, Maria Elena Vega Sanchez.

**Supervision:** Jan De Backer, Rohit Gupta, Fredric Jaffe, Maria Elena Vega Sanchez.

**Writing – original draft:** Daniel Salerno, Ifeoma Oriaku, Melinda Darnell, Maarten Lanclus, Jan De Backer, Victor Kim.

**Writing – review & editing:** Daniel Salerno, Ben Lavon, Victor Kim.

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
