## [Decision Letter · Decision Letter 0]

11 Aug 2021

PONE-D-21-15180

Association of Abnormal Pulmonary Vasculature on CT Scan for COVID-19 Infection with Decreased Diffusion Capacity in Follow Up: a retrospective cohort study.

PLOS ONE

Dear Dr. Salerno,

Thank you for submitting your manuscript to PLOS ONE. After careful consideration, we feel that it has merit but does not fully meet PLOS ONE’s publication criteria as it currently stands. Therefore, we invite you to submit a revised version of the manuscript that addresses the points raised during the review process.

The topic addressed in the present paper is interesting on a clinical point of view. However, the sample size is small and this deserves the assessment of the statistical power, in order to make results useful for other centers. All factors should be considered in the multivariable analysis. 

We look forward to receiving your revised manuscript.

Kind regards,

Chiara Lazzeri

Academic Editor

PLOS ONE

1. Please ensure that your manuscript meets PLOS ONE's style requirements, including those for file naming. The PLOS ONE style templates can be found at https://journals.plos.org/plosone/s/file?id=wjVg/PLOSOne_formatting_sample_main_body.pdf and https://journals.plos.org/plosone/s/file?id=ba62/PLOSOne_formatting_sample_title_authors_affiliations.pdf.

4. Thank you for stating the following in the Acknowledgments/Funding Section of your manuscript:

“Image analyses were funded by Fluidda’s COVID19 consortium”

“Image analyses were funded by Fluidda’s COVID19 consortium. The funders had no role in study design, data collection and analysis, decision to publish, or preparation of the manuscript.”

“Over the past three years, VK reports personal fees from Gala Therapeutics, the ABIM, AstraZeneca, and Boehringer Ingelheim outside the submitted work; ML, JDB, BL have received monies from Fluidda, Inc; IO, MD, RG, FJ, MEVS, DS have no competing interests to declare.”

7. We note that Figure 2 in your submission contain copyrighted images. All PLOS content is published under the Creative Commons Attribution License (CC BY 4.0), which means that the manuscript, images, and Supporting Information files will be freely available online, and any third party is permitted to access, download, copy, distribute, and use these materials in any way, even commercially, with proper attribution. For more information, see our copyright guidelines: http://journals.plos.org/plosone/s/licenses-and-copyright.

Additional Editor Comments (if provided):

Reviewers' comments:

Reviewer's Responses to Questions

**Comments to the Author**

1. Is the manuscript technically sound, and do the data support the conclusions?

Reviewer #1: Yes

2. Has the statistical analysis been performed appropriately and rigorously? 

Reviewer #1: Yes

3. Have the authors made all data underlying the findings in their manuscript fully available?

Reviewer #1: Yes

4. Is the manuscript presented in an intelligible fashion and written in standard English?

Reviewer #1: Yes

5. Review Comments to the Author

Reviewer #1: Through an experimental design close to clinical practice, the author retrospectively analyzed the possible correlation with abnormal pulmonary function tests after COVID-19 from a unique perspective, and the conclusion is helpful for the clinical diagnosis and treatment of COVID-19.

As the author himself stated in the article, this article has certain deficiencies:

1. The sample size is small, and the analysis and conclusions of the data are scientifically limited.

2. Factors related to analysis can be more adequate, such as analysis of pulmonary vascular injury and microvascular thrombosis etc.

In any case, the author proposes a unique perspective to analyze the clinical manifestations of COVID-19, which can be studied in more depth.

6. PLOS authors have the option to publish the peer review history of their article (what does this mean?). If published, this will include your full peer review and any attached files.

Reviewer #1: **Yes: **Haimei MA

---

## [Author Response · Author response to Decision Letter 0]

5 Sep 2021

1) Response to Academic Editor:

Thank you for your insightful review of our manuscript. We agree that the sample size is small and it would be difficult to extrapolate the numbers needed to analyze in order to make a more definitive conclusion about abnormal pulmonary vasculature and risk or developing a reduced DLCO. Could you elaborate? Would you like us to make a more definitive statement about the statistical power in the manuscript? Additionally, we would like you to elaborate on the statement “all factors should be considered in the multivariable analysis.” The following variables were used as covariates: age, sex, race, smoking status, lung volumes, and hemoglobin. In order to address this statement, could you be more specific?

2) Response to Reviewer #1:

Thank you for your insightful review of our manuscript. We completely agree with your assessment. We added in our limitations the following to reflect into your comments: Sixth, we did not have in our cohort other test (such as ventilation-perfusions scans, CT pulmonary angiography, dual energy CT) that could have provided more insight into pulmonary vascular injury related to COVID-19 pulmonary disease.

We also did some minor updates in the background and discussion section, we added 3 new references. We updated current number of COVID 19 cases and deaths in the US in the background section.

---

## [Editor Report · Decision Letter 1]

14 Sep 2021

Association of Abnormal Pulmonary Vasculature on CT Scan for COVID-19 Infection with Decreased Diffusion Capacity in Follow Up: a retrospective cohort study.

PONE-D-21-15180R1

Dear Dr. Salerno,

We’re pleased to inform you that your manuscript has been judged scientifically suitable for publication and will be formally accepted for publication once it meets all outstanding technical requirements.

Kind regards,

Chiara Lazzeri

Academic Editor

PLOS ONE
---

## [Editor Report · Acceptance letter]

8 Oct 2021

PONE-D-21-15180R1 

Association of Abnormal Pulmonary Vasculature on CT Scan for COVID-19 Infection with Decreased Diffusion Capacity in Follow Up: a retrospective cohort study. 

Dear Dr. Salerno:

I'm pleased to inform you that your manuscript has been deemed suitable for publication in PLOS ONE. Congratulations! Your manuscript is now with our production department. 

Kind regards, 

on behalf of

Dr. Chiara Lazzeri 

Academic Editor

PLOS ONE